# In-Hand Object Rotation via Rapid Motor Adaptation

**Haozhi Qi**[*,1,2]    **Ashish Kumar**[*,1]    **Roberto Calandra**[2]    **Yi Ma**[1]    **Jitendra Malik**[1,2]

[1]UC Berkeley    [2]Meta AI

https://haozhi.io/hora/

Train in Simulation      Directly Deploy in the Real World

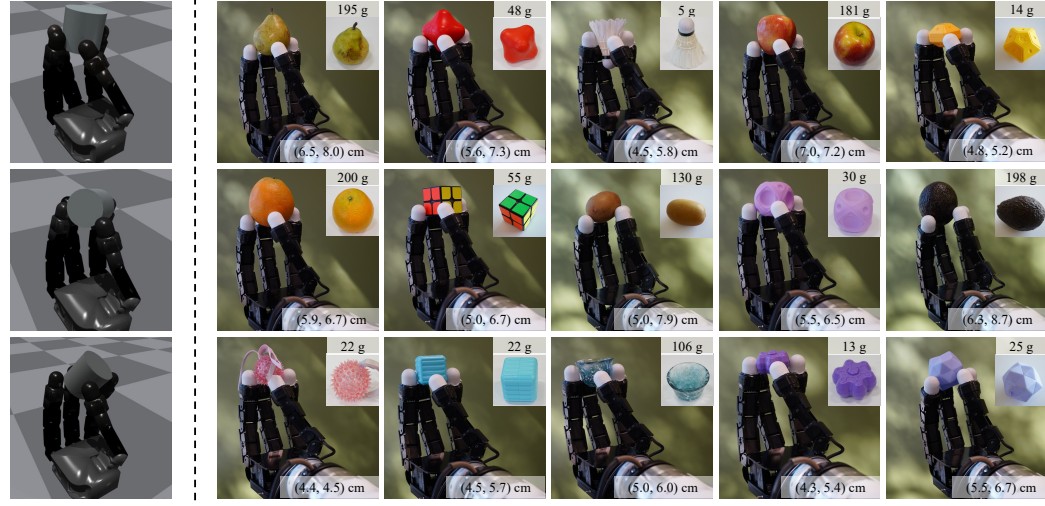

Figure 1: **Left:** Our controller is trained only in simulation on simple cylindrical objects of different sizes and weights. **Right:** Without any real world fine-tuning, the controller can be deployed to a real robot on a diverse set of objects with different shapes, sizes and weights (object mass and the shortest/longest diameter axis length along the fingertips are shown in the figure) using only proprioceptive information. **Website**: Emergence of natural stable finger gaits can be observed in the learned control policy.

**Abstract:** Generalized in-hand manipulation has long been an unsolved challenge of robotics. As a small step towards this grand goal, we demonstrate how to design and learn a simple adaptive controller to achieve in-hand object rotation using only fingertips. The controller is trained entirely in simulation on only cylindrical objects, which then – without any fine-tuning – can be directly deployed to a real robot hand to rotate dozens of objects with diverse sizes, shapes, and weights over the $z$-axis. This is achieved via rapid online adaptation of the robot's controller to the object properties using only proprioception history. Furthermore, natural and stable finger gaits automatically emerge from training the control policy via reinforcement learning. Code and more videos are available at our Website.

**Keywords:** In-Hand Manipulation, Object Rotation, Reinforcement Learning

## 1   Introduction

Humans are remarkably good at manipulating objects in-hand – they can even adapt to new objects of different shapes, sizes, mass and materials with no apparent effort. While several works have shown in-hand object rotation with real-world multi-fingered hands for a single or a few objects [1, 2, 3, 4], truly generalizable in-hand manipulation remains an unsolved challenge of robotics.

In this paper, we demonstrate that it is possible to train an adaptive controller capable of rotating diverse objects over the $z$-axis with the fingertips of a multi-fingered robot hand (Figure 1). This task is a simplification of the general in-hand reorientation task, yet still quite challenging for robots since at all times the fingers need to maintain a dynamic or static force closure on the object to prevent it from falling (as it can not make use of any other supporting surface such as the palm).

---

[*]Equal Contribution.

6th Conference on Robot Learning (CoRL 2022), Auckland, New Zealand.

Our approach is inspired by the recent advances in legged locomotion [5, 6] using reinforcement learning. The core of these works is to learn a compressed representation of different terrain properties (called *extrinsics*) for walking, which is jointly trained with the control policy. During deployment, the extrinsics is estimated online and the controller can perform rapid adaptation to it. Our key insight is that, despite the diversity of real-world objects, for the task of in-hand object rotation, the important physical properties such as local shape, mass, and size as perceived by the *fingertips* can be compressed to a compact representation. Once such a compressed representation (*extrinsics*) of different objects is learned, the controller can estimate it online from proprioception history and use it to adaptively manipulate a diverse set of objects.

Specifically, we encode the object's intrinsic properties (such as mass and size) to an extrinsics vector, and train an adaptive policy with it as an input. The learned policy can robustly and efficiently rotate different objects in simulation environments. However, we do not have access to the extrinsics when we deploy the policy in the real world. To tackle this problem, we use the rapid motor adaptation [6] to learn an adaptation module which estimate the extrinsic vector, using the discrepancy between observed proprioception history and the commanded actions. This adaptation module can also be trained solely in simulation via supervised learning. The concept of estimating physical properties using proprioceptive history has been widely used in locomotion [5, 6, 7] but has not yet been explored for in-hand manipulation.

Experimental results on a multi-finger Allegro Hand [8] show that our method can successfully rotate over 30 objects of diverse shapes, sizes (from $4.5\,\mathrm{cm}$ to $7.5\,\mathrm{cm}$), mass (from $5\,\mathrm{g}$ to $200\,\mathrm{g}$), and other physical properties (e.g., deformable or soft objects) in the real world. We also observe that an adaptive and smooth finger gait emerges from the learning process. Our approach shows the surprising effectiveness of using only proprioception sensory signals for adaptation to different objects, even without the usage of vision and tactile sensing. To further understand the underlying mechanisms of our approach, we studied the estimated extrinsics when manipulating different objects. We find interpretable extrinsic values that correlate to mass and scale changes, and that a low-dimensional structure of the embedding does exist, both of which are critical to our generalization ability.

## 2  Related Work

**Classic Control for In-Hand Manipulation.** Dexterous in-hand manipulation has been an active research area for decades [9]. Classic control methods usually need an analytical model of the object and robot geometry to perform motion planning for object manipulation. For example, [10, 11] rely on such a model to plan finger movement to rotate objects. [12] assumes objects are piece-wise smooth and use finger-tracking to rotate objects. [13, 14] demonstrate reorientation of different objects in simulation by generating trajectories using optimization. There have been also attempts to deploy systems in the real-world. For example, [15] calculates precise contact locations to plan a sequence of contact locations for twirling objects. [16] plans over a set of predefined grasp strategies to achieve object reorientation using two multi-fingered hands. [17, 18] use throwing or external forces to perturb the object in the air and re-grasp it. Recently, works such as [19, 20] do in-grasp manipulation without breaking the contact. [4] demonstrates complex object reorientation skills using a non-anthropomorphic hand by leveraging the compliance and an accurate pose tracker. The diversity of the objects they can manipulate is still limited due to the intrinsic complexity of the physical world. In contrast to traditional control approaches which may use heuristics or simplified models to solve this task, we instead use model-free reinforcement learning to train an adaptive policy, and use adaptation to achieve generalization.

**Reinforcement Learning for In-Hand Manipulation.** To get around the need of an accurate object model and physical property measures, in the last few years, there has been a growing interest in using reinforcement learning directly in the real-world for dexterous in-hand manipulation. [21] learns simple in-grasp rolling for cylindrical objects. [22, 23] learns a dynamics model and plan over it for rotating objects on the palm. [24, 25] use human demonstration to accelerate the learning process. However, since reinforcement learning is very sample inefficient, the learned skills are rather simple or have limited object diversity. Although complex skills such as re-orientating a diverse set of objects [26, 27, 28] and tool use [29, 30] can be obtained in simulation, transferring the results to real-world remains challenging. Instead of directly training a policy in the real world, our approach learns the policy entirely in the simulator and aims to directly transfer to the real world.

**Sim-to-Real Transfer via Domain Randomization.** Several works aim to train reinforcement learning policies using a simulator and directly deploy it in a real-world system. Domain randomiza-

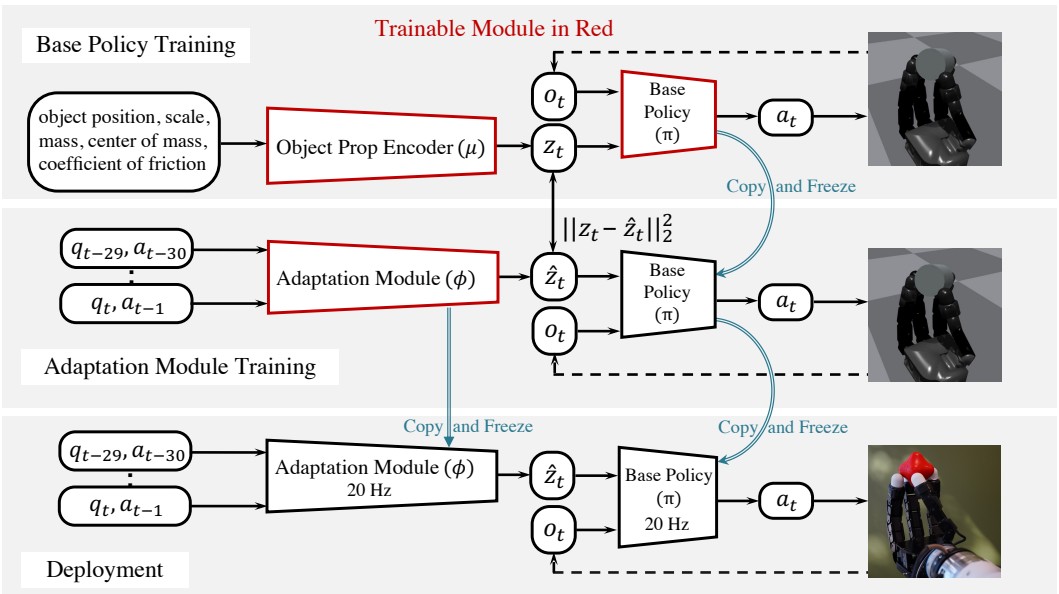

Figure 2: An overview of our approach at different training and deployment stages. In *Base Policy Learning*, we jointly optimize $\mu$ and $\pi$ using PPO [33]. The observation $o_t$ only contains three past joint positions and commanded actions. Next, in *Adaptation Module Learning*, we freeze the policy $\pi$ and use supervised learning to train $\phi$ which uses proprioception and action history to estimate the extrinsics vector $z_t$. During *Deployment*, the base policy $\pi$ uses the extrinsics $\hat{z}_t$ estimated and updated online by $\phi$.

tion [31] varies the simulation parameters during training to expose the policy to diverse simulation environments so that they can be robustly deployed in the real-world. Representative examples are [1] and [2]. They leverage massive computational resources and large-scale reinforcement learning methods to learn an agile object reorientation skills and solve Rubik's Cube with a single robot hand. However, they still focus only on manipulating a limited number of objects. [3] learns a finger-gaiting behavior efficiently and transfers to a real robot when hand facing downwards, but they do not only use fingertips and the objects considered are all cubes. Our approach focuses on generalization to a diverse set of objects and can be trained within a few hours.

**Sim-to-Real via Adaptation.** Instead of relying on domain randomization which is agnostic to current environment parameters, [32] performs system identification via initial calibration, or online adaptive control to estimate the system parameters for Sim-to-Real Transfer. However, learning the exact physical values and alignment between simulation and real-world may be sub-optimal because of the intrinsic inaccuracy of physics simulations. An alternative way is to learn a low dimensional embedding which encodes the environment parameters [5, 6] which is then used by the control policy to act. This paradigm has enabled robust and adaptive locomotion policies. However, it is not straightforward to apply it on the in-hand manipulation task. Our approach demonstrates how to design the reward and training environment to enable a natural and stable controller that can transfer to the real world.

## 3 Rapid Motor Adaptation for In-Hand Object Rotation

An overview of our approach is shown in Figure 2. During deployment (Figure 2, bottom), our policy infers a low-dimensional embedding of object's properties such as size and mass from proprioception and action history, which is then used by our base policy to rotate the object. We first describe how we train a *base policy* with object property provided by a simulator, then we discuss how to train an *adaptation module* that is capable of inferring these properties.

### 3.1 Base Policy Training

**Privileged Information.** In this paper, privileged information refers to the object's properties such as position, size, mass, coefficient of friction, center of mass of the object. This information, denoted by a 9-dim vector $e_t \in \mathbb{R}^9$ at timestep $t$, can be accurately measured in simulation. We provide

this as an input to the policy, but instead of using $e_t$ directly, we use an 8-dim embedding (called extrinsics in [6]) $z_t = \mu(e_t)$ which gives better generalization behavior as we show in Section 5.

**Base Policy.** Our control policy $\pi$ takes the current robot joint positions $q_t \in \mathbb{R}^{16}$, the predicted action $a_{t-1} \in \mathbb{R}^{16}$ at the last timestep, together with the extrinsics vector $z_t \in \mathbb{R}^8$ as input, and outputs the target of the PD Controller (denoted as $a_t$). We also augment the observation to include two additional timesteps to have the velocity and acceleration information. Formally, the base policy output $a_t = \pi(o_t, z_t)$ where $o_t = (q_{t-2:t}, a_{t-3:t-1}) \in \mathbb{R}^{96}$.

**Reward Function.** We jointly optimize the policy $\pi$ and the embedding $\mu$ using PPO [33]. The reward function depends on several quantities: $\omega$ is the object's angular velocity. $\hat{k}$ is the desired rotation axis (we use the $z$-axis in the hand coordinate system). $q_{\text{init}}$ is the starting robot configuration. $\tau$ is the commanded torques at each timestep. $v$ is the object's linear velocity. Our reward function $r$ (subscript $t$ omitted for simplicity) to maximize is then

$$r \doteq r_{\text{rot}} + \lambda_{\text{pose}}\, r_{\text{pose}} + \lambda_{\text{linvel}}\, r_{\text{linvel}} + \lambda_{\text{work}}\, r_{\text{work}} + \lambda_{\text{torque}}\, r_{\text{torque}} \tag{1}$$

where $r_{\text{rot}} \doteq \max(\min(\omega \cdot \hat{k}, r_{\max}), r_{\min})$ is the rotation reward, $r_{\text{pose}} \doteq -\|q - q_{\text{init}}\|_2^2$ is the hand pose deviation penalty, $r_{\text{torque}} \doteq -\|\tau\|_2^2$ is the torque penalty, $r_{\text{work}} \doteq -\tau^T \dot{q}$ is the energy consumption penalty, and $r_{\text{linvel}} \doteq -\|v\|_2^2$ is the object linear velocity penalty. Note that in contrast to [28] which explicitly encourages at least three fingertips to always be in contact with the object, we do not enforce any heuristic finger gaiting behaviour. Instead, a stable finger gaiting behaviour emerges from the energy constraints and the penalty on deviation from the initial pose.

**Object Initialization and Dynamics Randomization.** A good training environment has to provide enough variety in simulation to enable generalization in the real world. In this work we find that using cylinders with different aspect ratios and masses provides such variety. We uniformly sample different diameters and side lengths of the cylinder.

We initialize the object and the fingers in a stable precision grasp. Instead of constructing the fingertip positions as in [28], we simply randomly sample the object position, pose, and robot joint position around a canonical grasp until a stable grasp is achieved. We also randomize the mass, center of mass, and friction of these objects (see appendix for the details).

### 3.2 Adaptation Module Training

We cannot directly deploy the learned policy $\pi$ to the real world because we do not directly observe the vector $e_t$ and hence we cannot compute the extrinsics $z_t$. Instead, we estimate the extrinsics vector $\hat{z}_t$ from the discrepancy between the proprioception history and the commanded action history via an adaptation module $\phi$. This idea is inspired by recent work in locomotion [5, 6] where the proprioception history is used to estimate the terrain properties. We show that this information can also be used to estimate the object properties.

To train this network, we first collect trajectories and privileged information by executing the policy $a_t = \pi(o_t, \hat{z}_t)$ with the predicted extrinsic vectors $\hat{z}_t = \phi(q_{t-k:t}, a_{t-k-1:t-1})$. Meanwhile we also store the ground-truth extrinsic vector $z_t$ and construct a training set

$$\mathcal{B} = \{(q_{t-k:t}^{(i)}, a_{t-k-1:t-1}^{(i)}, z_t^{(i)}, \hat{z}_t^{(i)})\}_{i=1}^N.$$

Then we optimize $\phi$ by minimizing the $\ell_2$ distance between $z_t$ and $\hat{z}_t$ using Adam [34]. The process is iterative until the loss converges. We apply the same object initialization and dynamics randomization setting as the above section.

## 4 Experimental Setup and Implementation Details

**Hardware Setup.** We use an Allegro Hand from Wonik Robotics [8]. The Allegro hand is a dexterous anthropomorphic robot hand with four fingers, with each finger having four degrees of freedom. These 16 joints are controlled using position control at $20\,\text{Hz}$. The target position commands are converted to torque using a PD Controller ($K_p = 3.0$, $K_d = 0.1$) at $300\,\text{Hz}$.

**Simulation Setup.** We use the IsaacGym [35] simulator. During training, we use 16384 parallel environments to collection samples for training the agent. Each environment contains a simulated Allegro Hand and an cylindrical object with different shape and physical properties (the exact parameters are in the supplementary material). The simulation frequency is $120\,\text{Hz}$ and the control frequency is $20\,\text{Hz}$. Each episode lasts for 400 control steps (equivalent to $20\,\text{s}$).

| Method | Within Training Distribution | | | | Out-of-Distribution | | | |
|---|---|---|---|---|---|---|---|---|
| | RotR ($\uparrow$) | TTF ($\uparrow$) | ObjVel ($\downarrow$) | Torque ($\downarrow$) | RotR ($\uparrow$) | TTF ($\uparrow$) | ObjVel ($\downarrow$) | Torque ($\downarrow$) |
| Expert | $233.71_{\pm25.24}$ | $0.85_{\pm0.01}$ | $0.28_{\pm0.05}$ | $1.24_{\pm0.19}$ | $165.07_{\pm15.65}$ | $0.71_{\pm0.04}$ | $0.42_{\pm0.06}$ | $1.24_{\pm0.16}$ |
| Periodic | $43.62_{\pm2.52}$ | $0.44_{\pm0.12}$ | $0.72_{\pm0.21}$ | $1.77_{\pm0.49}$ | $22.45_{\pm0.59}$ | $0.34_{\pm0.08}$ | $1.11_{\pm0.19}$ | $1.41_{\pm0.54}$ |
| NoAdapt | $90.89_{\pm4.85}$ | $0.65_{\pm0.07}$ | $0.44_{\pm0.11}$ | $1.34_{\pm0.12}$ | $54.50_{\pm3.91}$ | $0.51_{\pm0.06}$ | $0.63_{\pm0.13}$ | $1.34_{\pm0.11}$ |
| DR | $176.12_{\pm26.47}$ | $0.81_{\pm0.02}$ | $0.34_{\pm0.05}$ | $1.42_{\pm0.06}$ | $140.80_{\pm17.51}$ | $0.63_{\pm0.02}$ | $0.64_{\pm0.06}$ | $1.48_{\pm0.20}$ |
| SysID | $174.42_{\pm23.31}$ | $0.81_{\pm0.02}$ | $0.32_{\pm0.03}$ | $1.29_{\pm0.72}$ | $132.56_{\pm17.42}$ | $0.62_{\pm0.09}$ | $0.50_{\pm0.09}$ | $1.26_{\pm0.17}$ |
| Ours | $\mathbf{222.27_{\pm21.20}}$ | $\mathbf{0.82_{\pm0.02}}$ | $\mathbf{0.29_{\pm0.05}}$ | $\mathbf{1.20_{\pm0.19}}$ | $\mathbf{160.60_{\pm10.22}}$ | $\mathbf{0.68_{\pm0.07}}$ | $\mathbf{0.47_{\pm0.07}}$ | $\mathbf{1.20_{\pm0.17}}$ |

Table 1: We compare our method to several baselines in simulation in two settings: 1) *Within Training Distribution*; 2) *Out-of-Distribution*. Our method with online continuously adaptation achieves the best performance compared to all the baselines, closely emulating the performance of the Expert that has the privileged information as the input.

**Baselines.** We compare our method to the baselines listed below. We also compare with the policy with access to privileged information (*Expert*), as the upper bound of our method (Figure 2, top row).

1. *A Robust Policy trained with Domain Randomization (DR):* This baseline is trained with the same reward function but without privileged information. This gives a policy which is robust, instead of adaptive, to all the shape and physical property variations [1, 3, 2, 36].
2. *Online Explicit System Identification (SysID):* This baseline predicts the exact system parameters $e_t$ instead of the extrinsic vector $z_t$ during training the adaptation module.
3. *No Online Adaptation (NoAdapt):* During deployment, the extrinsics vector $\hat{z}_t$ is estimated at the first time step and stays frozen during the rest of the run. This is to study the importance of online adaptation enabled by the adaptation module $\phi$.
4. *Action Replay (Periodic):* We record a reference trajectory from the expert policy with privileged information and run it blindly. This is to show our policy adapts to different objects and disturbances instead of periodically executing the same action sequence.

**Metrics.** We use the following metrics to compare the performance of our method to baselines.

1. *Time-to-Fall (TTF).* The average length of the episode before the object falls out of the hand. This value is normalized by the maximum episode length (20s in simulation experiments and 30s in the real world experiments).
2. *Rotation Reward (RotR).* This is the average rotation reward ($\boldsymbol{\omega} \cdot \hat{\boldsymbol{k}}$) of an episode in simulation. Note that we do not train with this reward. Instead, we use a clipped version of this reward during training.
3. *Radians Rotated within Episode (Rotations).* Since angular velocity of the object is hard to accurately measure in the real world, we instead measure the net rotation of the object (in radians) achieved by the policy with respect to the world's z-axis. This metric is only used in the real world experiments.
4. *Object's Linear Velocity (ObjVel).* We measure the magnitude of the linear velocity to measure the stability of the object. This is only measured in simulation. The value is scaled by 100.
5. *Torque Penalty (Torque).* We measure the average $\ell_1$ norm of commanded torque per timestep during execution to measure the energy efficiency.

## 5    Results and Analysis

In this section, we compare the performance of our method to several baselines both in simulation and in real-world deployment. We also analyze what the adaptation module learns and how it changes during policy execution and as the objects change. Finally, we focus on training a policy for rotating an object along the negative z-axis with respect to the world coordinate. We also explore the possibility of training a multi-axis policy ($\pm$ $z$-axis) in the appendix. More qualitative results of our method and several ablations of our method are in our Project Website and the supplementary.

### 5.1    Generalization via Adaptation

**Comparison in Simulation.** We first compare our method with the baselines mentioned in Section 4 in simulation. We evaluate all methods under two settings: 1) In the *Within Training Distribution*

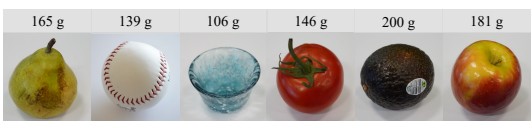

| Method | Rotations ($\uparrow$) | TTF ($\uparrow$) | Torque ($\downarrow$) |
|---|---|---|---|
| DR | $9.67_{\pm 4.33}$ | $0.72_{\pm 0.34}$ | $2.03_{\pm 0.36}$ |
| SysID | $10.36_{\pm 2.32}$ | $0.61_{\pm 0.33}$ | $1.88_{\pm 0.38}$ |
| NoAdapt | N.A. | $0.35_{\pm 0.20}$ | N.A. |
| Ours | $\mathbf{23.96}_{\pm 3.16}$ | $\mathbf{0.98}_{\pm 0.08}$ | $\mathbf{1.84}_{\pm 0.24}$ |

Figure 3: Quantitative Evaluation on a diverse set of Heavy Objects (Left). Our method which uses adaptation performs the best in terms of total rotated angle (in radians), Time To Fall (TTF), and Energy efficiency (Torque). The *DR* baseline has a conservative policy which results in a slower angular velocity. *SysID* has a more dynamic and agile policy but very unstable as can be seen in a lower TTF compared to *DR* and ours. The *NoAdapt* baseline fails on the task showing the importance of continuous online adaptation.

setting, we use the same object set and randomization setting as in RL training; 2) In the *Out-of-Distribution* setting, we use objects with a larger range of physical randomization range. We also change 20% of the objects to be spheres and cubes. We compute the average performance over 500K episodes with different parameter randomizations and initialization conditions. We report the average and standard deviation over five models trained with different seeds.

The results in Table 1 show that our method with online adaptation achieves the best performance compared to all the baselines. We see that adaptation to shape and dynamics of the object not only enables a better performance in training, but also gives a much better generalization to out-of-distribution object parameters compared to all the baseline methods. The *Periodic* baseline (i.e., simply playback of the expert policy) does not give a reasonable performance. Although it can rotate the exact same object with the same initial grasp and dynamic parameters, it fails to generalize beyond this very narrow setup. This baseline helps us understand the difficulty of the problem. The *NoAdapt* also performs poorly compared to our method which uses continuous online adaptation. The weaker performance of this baseline is explained by the fact that it does not update the extrinsics during the episode. This shows the importance of continuous online adaptation. The *DR* baseline, although roughly matches or performs better than the other baselines in terms of *RotR* and *TTF*, it is worse in other metrics related to object stability and energy efficiency. This is because the *DR* baseline is unaware of the underlying object properties and needs to learn a single gait, instead of an adaptive one, for all possible objects. The *SysID* baseline also performs worse than our method in both of the evaluation distributions. This is because it is harder as well as unnecessary to learn the exact values of the shape and dynamics parameters to adapt. This comparison shows the benefit of learning a low-dimensional compact representation which has a coarse relative activation for different physical properties instead of the exact value (see Figure 5 and Figure 6).

**Real-World Comparisons.** Next, we show real world comparisons of our policy to the baselines on two set of challenging objects shown in Figure 3 and Figure 4. We exclude the *Periodic* baseline because it does not work well even in Simulation and *Expert* because we cannot access privileged information in the real world. We evaluate each method using 20 different initial grasping positions and 6 different objects in each set. The maximum episode length is $30\,\mathrm{s}$.

We first study the behavior of our policy and different baselines in a set of heavy objects (more than $100\,\mathrm{g}$) including a baseball, different fruits, vegetables, and a cup (Figure 3). Our method performs significantly better in all metrics. Our method can achieves consistent rotation for almost all trials without falling down with an average Rotations of 23.96 (with a equivalent rotation velocity $0.8\,\mathrm{rad\,s^{-1}}$). We find the *DR* baseline is very slow and conservative in all the trials since it is unaware of object properties and needs to learn a single gaiting behaviour for all objects. As a result, it has the lowest Rotations, although the TTF is higher than the *SysID* baseline. We also find the *DR* could perform reasonably well on objects of larger sizes because they are easier to rotate and forms a reasonable chunk of training distribution. The *SysID* baseline learns a more dynamic and agile behavior, having a better Rotations metric, but a lower TTF and Rotations showing that precise estimation of system parameters is both unnecessary and difficult. We find that it is particularly hard for it to generalize to the small cup and the slightly soft tomato. Lastly, we observe a similar behavior as in [6] that the *NoAdapt* baseline does not transfer successfully to the real world, showing the importance of continuous online adaptation for successful real-world deployment.

We perform the same comparisons on a collection of irregular objects (Figure 4). It contains a container with moving COM, objects with concavity, a cylindrical kiwi fruit, a shuttlecock, a toy with

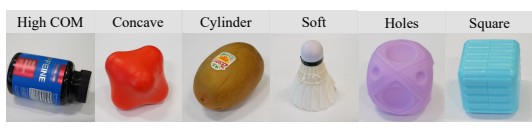

| Method | Rotations (↑) | TTF (↑) | Torque (↓) |
|---|---|---|---|
| DR | $6.59_{\pm3.71}$ | $0.66_{\pm0.41}$ | $1.85_{\pm0.37}$ |
| SysID | $8.16_{\pm3.39}$ | $0.46_{\pm0.36}$ | $1.70_{\pm0.40}$ |
| NoAdapt | N.A. | $0.12_{\pm0.05}$ | N.A. |
| Ours | $\mathbf{19.22_{\pm4.08}}$ | $\mathbf{0.78_{\pm0.27}}$ | $\mathbf{1.48_{\pm0.30}}$ |

Figure 4: Quantitative Evaluation on a diverse set of Irregular Objects (Left). Our method can successfully generalize to rotating a diverse set of objects including objects with holes, soft and deformable objects (none of these were included in the training). Our method outperforms the baselines on all the metrics. The *DR* baseline has the second highest TTF but low Rotations because it outputs very conservative and slow trajectories. *SysID* achieves slightly faster but a very unstable policy. The superior performance of our method over baselines shows the importance of adaptation via a low dimensional extrinsics estimation for generalization in this task.

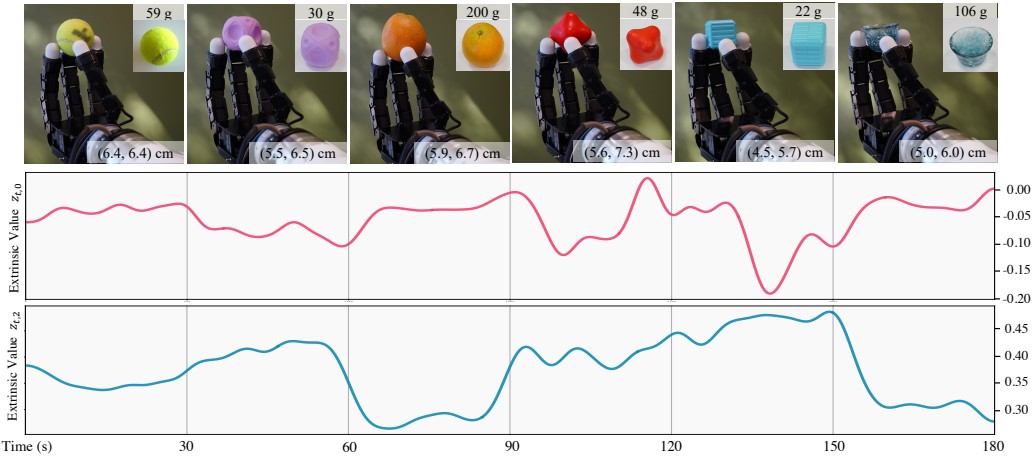

Figure 5: Two of the 8-dim estimated extrinsics vector during one continuous run in which we change the object in the hand every 30s for 6 objects. The top plot shows the extrinsic value $z_{t,0}$ which responds to changes in object diameter with smaller diameters leading to lower values. Since the perceived diameter of the object changes during rotating an irregular shaped object, we see variations in this extrinsic value even within the same object. The bottom plot shows the correlation between $z_{t,2}$ and object mass: it attains higher values for lighter objects and lower for heavier objects. See this link for the video.

holes, and a cube toy. Rotating the cube is particularly difficult because we only rely on usage of fingertips. Although these variations are challenging for this task and are beyond what is seen during training, our method still performs reasonably well, outperforming all the baselines. We see that the *DR* baseline can perform stable but slow in-hand rotation for the container and the shuttlecock, but largely fails for the other objects, indicating its difficulty in shape generalization. For the *SysID* baseline, its stability is significantly lower than the *DR* baseline as well as our method, despite having a higher angular velocity. The *NoAdapt* baseline performs similarly to what we observed in Figure 3.

## 5.2  Understanding and Analysis

To understand how our method generalizes to a diverse set of objects, we design a few experiments and visualization to help develop some insights.

**Extrinsics over Time.** We run one continuous evaluation episode in the real world in which we replace the object in the hand every 30s for 6 objects. Note that during training, we never randomize the objects within an episode. During the entire run, we record the estimated extrinsics and plot 2 out of the 8-dim extrinsics vector in Figure 5. The top plot shows the extrinsic value $z_{t,0}$ which responds to changes in object diameter. It has a lower value for smaller diameters and higher value for larger diameters. The bottom plot shows that the extrinsic value $z_{t,2}$ responds to variations in object mass.

**Extrinsic Clustering.** Another way to understand the estimated extrinsics $\hat{z}$ is by clustering the extrinsics vector estimated while rotating different objects. In Figure 6, we visualize the estimated extrinsics vector using t-SNE for rotating 6 different objects. We find that objects of different sizes and different weights tend to occupy separate regions as explained in Figure 6.

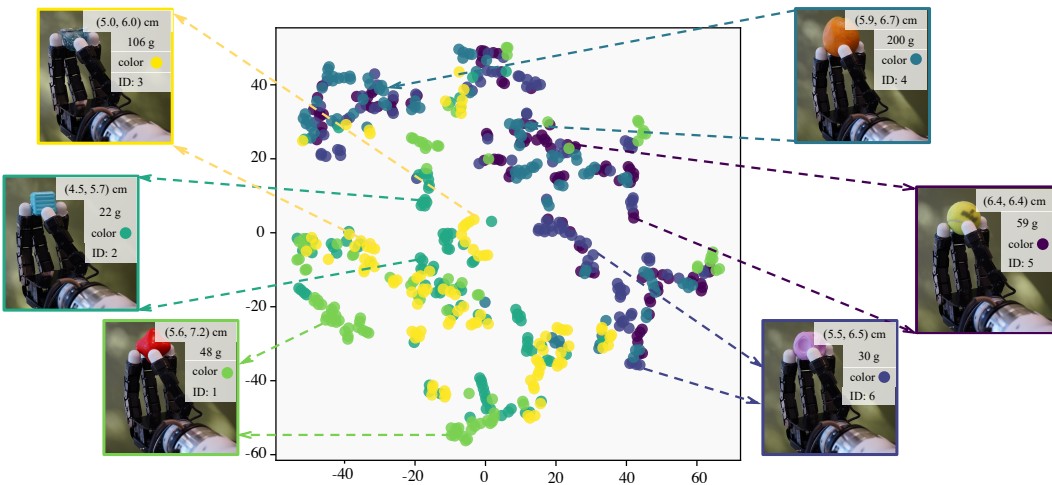

Figure 6: We visualize the estimated extrinsics vector $\hat{z}_t$ using t-SNE for rotating 6 different objects for 5 seconds each. We found that objects of different sizes and different weights tend to occupy separate regions. For example, $\hat{z}$ corresponding to smaller objects tend to cluster on the bottom left. The object with ID 1 (shown in the bottom left in the above figure) produces scattered $\hat{z}$ because of its irregular shape, which results in an equivalently varying scale of the object. This locality leads to the scattering of the extrinsics across the t-SNE plot, and this locality is also what gives us generalization as evaluated in Table 1, Figure 3 and Figure 4.

**Emergent Finger Gaits.** We find it important to use cylindrical objects for the emergence of a stable and high-clearance gait. On our website, we compare the learned finger gaits when training with cylindrical objects and with pure spherical objects. The latter training scheme leads to a policy with a dynamic gait which works well on balls but fail to generalize to more complex objects.

### 5.3 Real World Qualitative Results

On our Project Website, we qualitatively compare our method with the baselines on 5 different objects and further evaluate our method on 30 objects of different shapes and physical properties, including porous objects and non-rigid objects. The diameter ranges from $4.5\,\text{cm}$ to $7.5\,\text{cm}$ and the mass ranges from $5\,\text{g}$ to $200\,\text{g}$. Note that we only use cylindrical objects of different sizes and aspect ratios during training and show shape generalization in the real world with the use of rapid adaptation. The key insight which allows this generalization is that the shape of the objects as perceived by the fingertips of the hand can be compressed to a low dimensional space. We estimate this low dimensional space from proprioceptive history, which allows us to generalize to objects which are seemingly different in the real world but might look similar in the extrinsics space.

## 6 Discussion and Limitations

There are different levels of difficulty for general dexterous in-hand manipulation. The task considered in this paper (in-hand object rotation over the $z$-axis) is a simplification of the general SO(3) reorientation problem. However, it is not a limiting simplification. With three policies for rotation along three principle axes, we can achieve rotating the object to any target pose. We view this task as an important future extension of our work.

We show the feasibility of purely proprioceptive in-hand object rotation for a diverse set of objects. We find most of our failure cases in the real-world experiments are due to incorrect contact points which leads to unstable force closure. Since our method only relies on the proprioceptive sensing, it is unaware of the precise contact positions between the objects and the fingertips. Another failure case is when the object to be rotated is tiny (with diameter less than $4.0\,\text{cm}$). In this case the fingers will frequently collide with each other, making the robot not able to maintain the grasp balance of the object. More extreme and sophisticated shapes are also harder to manipulate. For those more challenging tasks, it might be necessary to incorporate tactile or visual feedback.

We aim to study the generalization to the real world via adaptation, so we do not utilize real-world experience to improve our policy. Incorporating real-world data to improve our policy (e.g. by using meta-learning) will be an interesting and meaningful next step.

**Acknowledgments**

This research was supported as a BAIR Open Research Common Project with Meta. In addition, in their academic roles at UC Berkeley, Haozhi, Ashish, and Jitendra were supported in part by DARPA Machine Common Sense (MCS) and Haozhi and Yi by ONR (N00014-20-1-2002 and N00014-22-1-2102). We thank Tingfan Wu and Mike Lambeta for their generous help on the hardware setup, Xinru Yang for her help on recording real-world videos. We also thank Mike Lambeta, Yu Sun, Tingfan Wu, Huazhe Xu, and Xinru Yang for providing feedback on earlier versions of this project.

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
