# OpenReview forum: "In-Hand Object Rotation via Rapid Motor Adaptation"
_robot-learning.org/CoRL/2022/Conference — CoRL 2022 Poster_

### Official Review · Reviewer_W5ch · 2022-07-04

**Originality:** Very Good
**Technical Quality:** Very Good
**Clarity Of Presentation:** Very Good
**Impact:** 3

**Recommendation:**

Weak Accept: I recommend accepting the paper, but will not argue for my recommendation if the majority of other reviewers have a different opinion.

**Summary:**

This paper applies the training pipeline of Rapid Motor Adaptation to the setting of in-hand object rotation. Note that in this paper, the `in-hand object rotation` actually means the robotic hand keeps rotating the objects along the fixed $z$-axis, not in the full $SO(3)$ space. The results show that the allegro hand can quasi-statically rotate around 30 different objects. The real-world experiments look nice.

**Issues:**

Listed above.

**Quality Of The Limitations Section:**

Limitations are not well addressed

**Reviewer Expertise:**

5: The reviewer is absolutely certain that the evaluation is correct and very familiar with the relevant literature

**Robotics Focus:**

Sufficient demonstration on hardware

**Strengths And Weaknesses:**

Strenghts:

* The overall learning pipeline is simple and convincing.
* The real-world experiments are nice and show that the policies actually transfer.
* It's nice to see that policies trained with only rotating a cylinder can rotate other objects to some extent.


Weaknesses:

* The paper is a direct application of a published method to a new task. The method itself is unchanged from the prior works.
* One baseline that is missing is the ablation of the adaptation module. While the paper has a baseline called NoAdapt, I think there is another baseline missing. The aspect that's missing is whether we actually need a separation of an adaptation module and a policy module. The paper currently regresses the output of the adaptation module to the privileged state embedding. Is this necessary? In other words, can we just train a temporal network $\pi(x_{t-30}, a_{t-30}, ..., x_{t-1}, a_{t-1}, o_t)$ that takes as input the history of observation and outputs the action command (no explicit latent regression)?


Claims/Statements that need to be revised and made clear:
* The learned gait is a quasi-static gait and the rotation is only around $z$-axis. From the demo videos, it's safe to say that at any time, the object is in a relatively stable pose with multiple finger support. This is sufficient for rotating the objects around the $z$-axis, which is not a bad thing. I think this is also why the policy works for different objects. The fingers have good support on the object being rotated, the policy only needs to figure out when contact happens between the fingertips and object (which can be inferred from the history of observations) and then move the fingers one by one.  The point I wanna make is that this task is much easier than dynamic object reorientation in full $SO(3)$ space that many prior works explored. So I don't think it is appropriate to criticize prior works (on in-hand object manipulation) on the lack of generality or sim2real transfer without stating what this paper is actually doing. They are tackling in-hand manipulation tasks with different levels of difficulty.

* The paper claims that the hand can rotate both convex and non-convex objects around $z$-axis. However, this is misleading. It is true that the objects used in the paper have different shapes. But what matters is the local geometry on which the contacts happen. For example, the paper uses bottle, paper towel, hourglass, sake cup, etc. At first glance, these objects look different. But because the paper only trains the hand to rotate objects around the vertical axis, the hand only needs to touch a very small convex (or almost convex) contour of the objects. And we can also see that on these objects (paper towel, hourglass, sake cup, shuttlecock, etc.), the rotation axis is carefully chosen such that other geometry of the objects does not affect the rotation. And the fingertips almost always work on the convex part of the objects. In this sense, these different objects used in the paper actually look quite similar. In addition, considering the fingertips of the Allegro hand are big, so the local contact geometry differences on the objects won't affect the motion as much. Therefore, I think it is misleading to claim that the policy can rotate both convex and non-convex objects. This should also be clearly discussed in the Limitation section.

**Summary Of Recommendation:**

I think the paper is working on an important problem. Dexterous manipulation is challenging. As a starting point, the paper addresses the problem of rotating objects around $z$-axis in a quasi-static way. The results are nice even though the method itself is completely from another published paper. However, I would require the authors to revise some misleading claims in the paper to make the writing more accurate. I am happy to raise my score if the writing is improved.

---

> ### Author Response · Authors · 2022-08-19
> **Re: Reviewer W5ch**
>
> Thanks for your detailed review! We hope the following answers can address your concern.
>
> ---
>
> > “The paper is a direct application of a published method to a new task. The method itself is unchanged from the prior works.”
>
> **Q: Novelty and Contribution.**
>
> A: Please see Q1/A1 of the shared response.
>
> ---
>
> > “One baseline that is missing is the ablation of the adaptation module ... can we just train a temporal network π(xt−30,at−30,...,xt−1,at−1,ot) that takes as input the history of observation and outputs the action command (no explicit latent regression)?”
>
> **Q: Baseline using temporal network but without explicit latent regression.**
>
> A: We show the advantage of using a separate extrinsic vector in the appendix (Table 3). The DR-* removes the extrinsic vector prediction and directly uses proprioceptive history as input of the policy. We try an LSTM, or an MLP to process the input history. We found it is more difficult to optimize those policies, especially with longer input timesteps. We hypothesize that this is because jointly learning a universal policy for a large range of physical variations is sample inefficient and unstable, while decoupling policy learning (RL) and extrinsic estimation (Supervised Learning) into two stages makes it more effective.
>
> ---
>
> > “The learned gait is a quasi-static gait and the rotation is only around the z-axis...The point I wanna make is that this task is much easier than dynamic object reorientation in full SO(3) space that many prior works explored. So I don't think it is appropriate to criticize prior works (on in-hand object manipulation) on the lack of generality or sim2real transfer without stating what this paper is actually doing. They are tackling in-hand manipulation tasks with different levels of difficulty.”
>
> **Q: Quasi-static Gait and comments on related work.**
>
> A: We agree the learned gait is quasi-static. This is a desired solution for the system as this enables resisting the random additional perturbations applied during training. We will clarify this in the paper.
>
> We also agree this is a simpler task than general SO(3) object reorientation, as we discussed in Q2 of the shared response. We do not aim to criticize prior works. The contrast with existing works on in-hand reorientation is to show that instead of solving the problem in its full complexity in simulation or for a few objects, instead, we can solve a simpler problem, but focus on and show the generalization in the real world. The contrast is primarily to put the paper in the context of existing works. We will update the introduction, related work, and the limitation section to clarify these aspects.
>
> ---
>
> > “The paper claims that the hand can rotate both convex and non-convex objects around the z-axis.”
>
> **Q: Non-Convex Objects.**
>
> A: We will remove the convex/non-convex related claims in the paper. We originally wrote that because our policy can rotate objects with holes or the reaction ball which are locally non-convex. This is not a particularly important aspect.

---

> > ### Comment · Reviewer_W5ch · 2022-08-25
> > **Post-rebuttal**
> >
> > I would like first to thank the authors for their response. I look forward to reading the updated version of the paper with corrected claims as the authors promised. I just wanna comment on the following two points. I wanna see what the authors think about the second comment.
> >
> > **Regarding the novelty and contribution**:
> >
> > I would consider the contribution of a reward function and a training environment to be rather small. For reorientation, there have been many past works investigating the reward functions (many of them are in the "related works"). The reward function presented in the paper is not too different from prior works. Each term in the proposed reward function has been used in prior works. And there is no evidence in the paper that the proposed reward function is the only one that works. In fact, many prior works that use a different reward function also work. So I would say the proposed reward function is a very small contribution.
> >
> >
> > **Regarding the baseline using temporal network but without explicit latent regression."**
> >
> > Sorry that I cannot seem to see the appendix now. If I understand the authors' response correctly, we are talking about two different baselines. In the "Adaptation Module Training", can we just use the supervised learning loss on the actions directly instead of trying to regress the $z_t$? More specifically, we can still train a teacher policy using the privileged information as done in this paper. Next, we just directly train a student policy that takes the same input ($x_{t-30}, a_{t-30}, ..., x_{t-1}, a_{t-1}, o_t$) by behavior cloning loss (so the only loss is an MSE loss on the action output $||a_t^{student}-a_t^{teacher}||$. This is the simplest form of teacher-student training. So my question is that is regression loss on the action output not sufficient? Do we really need to have an explicit intermediate output that predicts the $z_t$? Why does regression loss on the $z_t$ lead to easier optimization? As the "Base Policy" is fixed in the student training. These two losses essentially give the same training effect.

---

> > > ### Author Response · Authors · 2022-08-26
> > > **Re: Post-rebuttal**
> > >
> > > We have attached the appendix pdf for your reference.
> > >
> > > **Q: Minimize z_t or a_t.**
> > >
> > > A: We believe that it is important to have a two stage approach:
> > >   1) Train a base policy which has an extra input based on privileged information.
> > >   2) Use proprioceptive history to estimate that extra input during deployment time.
> > >
> > > We want to clarify that we didn’t claim optimizing z_t is easier than optimizing a_t. We only claim that the two phase optimization is easier and more effective than doing them together, as we show in appendix (Table 3).
> > >
> > > There is a choice in the second stage of whether to train with regression loss on z, on a, or both (with a fixed base policy). As the reviewer suggested, optimizing the z_t is not the only way. In fact, we explored this design choice during the method development and found the performance to be very similar. In the following table, we show the rotation reward, MSE loss of z_t, and MSE loss of a_t. All these three design choices give a similar result. And it also shows that even by optimizing for the action loss only, the network learns a \hat{z}_t which matches the z_t (the encoded ground truth privileged information).
> > >
> > > | Training Loss                                   | RotR         | z_t Loss        | a_t Loss        |
> > > |-------------------------------------------------|--------------|-----------------|-----------------|
> > > | \|\| $\hat{z}_t$ - $z_t$ \|\|                          | 222.27±21.20 | 0.0259 ± 0.0017 | 0.0108 ± 0.0034 |
> > > | \|\| $\hat{a}_t$ - $a_t$ \|\|                          | 221.88±22.82 | 0.0275 ± 0.0012 | 0.0102 ± 0.0031 |
> > > | \|\| $\hat{z}_t$ - $z_t$ \|\| + \|\| $\hat{a}_t$ - $a_t$ \|\| | 222.30±22.87 | 0.0262 ± 0.0018 | 0.0106 ± 0.0032 |
> > >
> > > We choose to optimize z_t in the submission because it’s simple and we don’t need to run the teacher policy network (to get a_t) at every training iteration.

---

### Official Review · Reviewer_p2uD · 2022-07-27

**Originality:** Good
**Technical Quality:** Very Good
**Clarity Of Presentation:** Very Good
**Impact:** 4

**Recommendation:**

Weak Accept: I recommend accepting the paper, but will not argue for my recommendation if the majority of other reviewers have a different opinion.

**Summary:**

The paper presents a system that learns an adaptive controller for in-hand manipulation, trained in simulation and deployed on a real robotic hand. The proposed approach relies on reinforcement learning in simulation, and does not require retraining or fine-tuning on the real robot, thanks to an adaptation module. The proposed system also relied on the fact that objects' characteristics, such as their shape and mass, as perceived by the fingertips proprioception can be compressed into a low dimensional embedding. Such representation then allows generalisation to unseen objects.

**Issues:**

Discussion on the simulation-reality gap in the context of the presented experimental setup: which part of your perception and control stack are less robust?
Reward formulation: which terms have the most impact?
Policy: motivate choices.

Minor: typos p.6 "The NoAdapt also *has performs* poorly...", " Our method *can achieves* consistent"

**Quality Of The Limitations Section:**

Limitations are addressed clearly

**Reviewer Expertise:**

3: The reviewer is fairly confident that the evaluation is correct

**Robotics Focus:**

Sufficient demonstration on hardware

**Strengths And Weaknesses:**

The paper presents a solid execution of an important topic in robot manipulation, that is in-hand manipulation. In particular, the proposed solution generalises across objects and is shown to be reliable on a real robotic hand. The proposed solution is simple and clean, although it has limitations (mentioned by the authors as well), by being only relying on proprioception.
The paper is written and structure in a clear way, and it is pleasant to read. Figures, diagrams and videos are well made and help clarify the contributions.
The paper includes a clear description of the experimental setup, and implementation details are reported (also in the supplementary material submitted).
Experiments include a comprehensive set of ablations and comparisons, which make the contribution of the paper more solid.

What is the (quantitative) effect of the gap between simulation and real setups? While usually more relevant for visual-related features, the domain gap for proprioceptive measurements could be limited. Is this true in your case? what is the main discrepancy playing a role in your experiment? As your perception stack is not based on visual clues, the appearance of the object in simulation should not affect your transfer, however mass or COM could be more relevant in your setup. Which object characteristic posed the biggest challenge for the sim-to-real transfer using your method?
The reward definition proposed includes several terms that are also commonly used to regularise actions in RL: do you have ablation studies or empirical results to highlight which component is mostly affecting the final behaviour?
Can you motivate the choice of using 2 time steps for your $o_t= (x_{t−2:t}, a_{t−3:t−1})$ in your policy?

**Summary Of Recommendation:**

The paper presents a solid design, implementation and execution of a method to address in-hand manipulation on a real robot hand with a variety of different objects. Although the components of the proposed framework are individually not novel on their own, their combination and the results presented are significant and the analysis conducted is thorough.

---

> ### Author Response · Authors · 2022-08-19
> **Re: Reviewer p2uD**
>
> Thanks for your detailed review! We hope the following answers can address your concern.
>
> ---
>
> > “What is the (quantitative) effect of the gap between simulation and real setups? While usually more relevant for visual-related features, the domain gap for proprioceptive measurements could be limited. Is this true in your case?”
>
> **Q: Quantitative Gap Between Sim and Real.**
>
> A: You are right that the domain gap for proprioceptive features is smaller than visual features. To quantitatively study it, we measure the rotations for the tennis ball in simulation and in the real-world. The size and mass of the simulated tennis ball is set to be the same as the real one. This is to show the performance gap between sim and real. In the simulation, the policy achieves an average of 30rad rotations within 30 seconds and 25rad in the real-world. This suggests that there still exists a small yet not negligible gap, potentially due to the sensing noise of the joint positions.
>
> ---
>
> > “What is the main discrepancy playing a role in your experiment? As your perception stack is not based on visual clues, the appearance of the object in simulation should not affect your transfer, however mass or COM could be more relevant in your setup. Which object characteristic posed the biggest challenge for the sim-to-real transfer using your method?”
>
> **Q: The most challenging object characteristic for sim-to-real transfer.**
>
> A: We found irregular shapes and aspect ratios are especially challenging during the sim-to-real transfer. Irregular shaped objects are challenging because it is difficult to accurately simulate them during training. Objects with extreme aspect ratios (such as phones or pens) are also hard since it is hard to estimate that property purely from proprioceptive history.
>
> ---
>
> > “The reward definition proposed includes several terms that are also commonly used to regularize actions in RL: do you have ablation studies or empirical results to highlight which component is mostly affecting the final behavior?”
>
> **Q: Ablation of Reward Function.**
>
> A: We qualitatively discuss the effect of each term in Section C of the appendix (the paragraph named Discussion on Reward Choice). We further conduct ablation experiments to show the effect on each term.
>
> | Method                | RotR         | TTF       | ObjVel    | Torque    |
> |-----------------------|--------------|-----------|-----------|-----------|
> | remove pose penalty   | 131.37±10.84 | 0.85±0.04 | 0.41±0.03 | 2.05±1.06 |
> | remove energy penalty | 212.33±10.28 | 0.79±0.02 | 0.48±0.02 | 1.73±0.22 |
> | remove linvel penalty | 226.46±19.87 | 0.75±0.04 | 0.38±0.04 | 1.22±0.11 |
> | full reward           | 222.27±21.20 | 0.82±0.02 | 0.29±0.05 | 1.20±0.19 |
>
> Without the pose penalty, the policy performs worse because the learned gait is unnatural and it does not learn to break and establish new contact. Please see an example in this link (http://anonymouscorl2022.github.io/ObjRotation/rebuttal.html#Reward). The energy penalty and linear velocity penalty greatly decrease the commanded torque and object’s linear velocity. This helps encourage the policy to learn a more stable and smooth gait to solve the task, and empirically yields better sim-to-real performance.
>
> ---
>
> > “Can you motivate the choice of using 2 time steps for your ot=(xt−2:t,at−3:t−1) in your policy?”
>
> **Q: Choice of using 2 Timesteps.**
>
> A: It allows the policy to implicitly estimate things such as joint velocity. We do not directly use the velocity because we empirically find that the velocity encoders from hardware are not stable.

---

> ### Author Response · Authors · 2022-08-25
> **Follow up on the response**
>
> Thanks again for your time. We hope our previous response addressed your concerns. Please let us know if you have additional questions or need more clarifications that can help us improve the paper.

---

> ### Author Response · Authors · 2022-08-26
> **Appendix**
>
> We just realize viewing appendix is temporarily disabled, so we attached the appendix for your reference.

---

### Official Review · Reviewer_5UyE · 2022-07-30

**Originality:** Good
**Technical Quality:** Good
**Clarity Of Presentation:** Excellent
**Impact:** 3

**Recommendation:**

Weak Accept: I recommend accepting the paper, but will not argue for my recommendation if the majority of other reviewers have a different opinion.

**Summary:**

This paper presents an algorithm to achieve in-hand object rotation through finger-gaiting using linkage-based robot hand. In the base policy training phase, a base policy was trained jointly with an object property encoder using proximal policy optimization with a custom-defined reward function. The object property encoder takes privileged object information and outputs an extrinsics vector, which, combined with the current robot joint position and the previous predicted robot action, are then used as the input of the base policy to output the next robot action. The next phase trained an adaptation module to output an estimated extrinsics vector using the proprioception history as input, in order to lift the dependency to privileged object information during deployment. The algorithm was tested in real world experiments using an Allegro hand, in comparison with different baseslines.

**Issues:**

* The outcome shows a potentially narrower scope compared to what was claimed in the paper. Based on the videos from the project page, the proposed method only achieved 2D manipulation: even though the objects are 3D, the rotations are all around the vertical axis ([0, 0, 1] and [0, 0, -1]). It is recommended to either claim the correct scope of the problem or provide further demonstrations to support the claim of general reorientation.
* The effective 2D aspect ratio of all objects are very close to 1.  Although there are a few long objects present in the experiment, their cross sections (where the fingers are in contact) have aspect ratio close to 1. This is okay as long as properly addressed in the limitation section. What would happen if the objects being manipulated (or their cross sections where the fingers are in contact with) have more extreme aspect ratios?  What is limiting the object generalizability? It will be very interesting to see the extrinsics vector of an more extreme shape compared to the nice shapes.


**Quality Of The Limitations Section:**

Limitations are not well addressed

**Reviewer Expertise:**

5: The reviewer is absolutely certain that the evaluation is correct and very familiar with the relevant literature

**Robotics Focus:**

Sufficient demonstration on hardware

**Strengths And Weaknesses:**

**Strengths:**
* Very clear presentation, especially section 3 with the help of Fig. 2.
* Multiple real world experiments demonstrating the effectiveness of the proposed method.

**Weaknesses:**
* The demonstrated object rotation is 2D. This would have been fine if it was well addressed in the paper (either in the scope or in the limitation).
* Because the manipulation is 2D, some of the experimental objects do not really demonstrate generalizability, because it is the 2D cross section that matters. Objects of different shapes could have very close cross sections.
* Not sufficiently encode object geometry: there is no object geometry information encoded in the object property encoder because it was not used as input for training. This could also explain the fairly limited generalizability for 2D manipulation.

**Summary Of Recommendation:**

The authors are solving the general object reorientation problem. While the proposed method aims to show general 3D reorientation that can be generalized to different objects, the experiment results only demonstrated object reorientation around two opposing axes (essentially 2D reorientation). This also made the object generalization less impressive, because objects such as the badminton and the cup have similar 2D cross section as a sphere. However, the method is still valid and was executed very well even with the limited scope.

---

> ### Author Response · Authors · 2022-08-19
> **Re: Reviewer 5UyE**
>
> Thanks for your detailed review! We hope the following answers can address your concern.
>
> ---
>
> > “The demonstrated object rotation is 2D. This would have been fine if it was well addressed in the paper (either in the scope or in the limitation).”
>
> **Q: The demonstrated object rotation is 2D.**
>
> A: We discuss the difficulty of the chosen task in Q2 of the shared response. We agree there are different difficulty levels for dexterous in-hand manipulation and our method is a small step towards that goal (written in the abstract). We will clearly clarify the difficulty level of the considered task in the revision.
>
> ---
>
> > “Because the manipulation is 2D, some of the experimental objects do not really demonstrate generalizability, because it is the 2D cross section that matters. Objects of different shapes could have very close cross sections.”
>
> **Q: Cross Section Matters.**
>
> A: The set of objects we choose have variations in physical properties including mass, center of mass, and how deformable they are. For object shape, we agree with the reviewer’s observation and actually utilize this fact to achieve generalization to different objects. As we write in the introduction, “Our key insight is that physical properties … as perceived by the fingertips can be compressed to a low-dimensional space. Despite the diversity of real-world objects … important factors can be defined by a few intrinsic properties of the object.” You are right that our method works with different objects with similar extrinsic embeddings. It is much more difficult for objects outside this distribution (tiny objects, objects with irregular aspect ratios), as we show in Table 5, 6, and 7 of the appendix.
>
> Meanwhile, the cross section is not the only important factor. For example, it is hard to infer an object's mass only from the cross section. Our policy infers the mass properties and behaves differently. We create a figure to show the relationship between object mass and commanded torque on the following link (https://anonymouscorl2022.github.io/ObjRotation/rebuttal.html#R2Q1).
>
> > “Not sufficiently encoded object geometry: there is no object geometry information encoded in the object property encoder because it was not used as input for training. This could also explain the fairly limited generalizability for 2D manipulation.”
>
> **Q: Object Geometry is not encoded.**
>
> A: You are right that we do not encode object geometry, which limits our generalization ability. In this project, we attempt to identify the simplest setup which can generalize to diverse objects in the real world. More sophisticated privileged information can potentially allow us to improve the performance, which we plan to do in the future. We will add this discussion in the limitation section.
>
> ---
>
> > “The outcome shows a potentially narrower scope compared to what was claimed in the paper. Based on the videos from the project page, the proposed method only achieved 2D manipulation: even though the objects are 3D, the rotations are all around the vertical axis ([0, 0, 1] and [0, 0, -1]). It is recommended to either claim the correct scope of the problem or provide further demonstrations to support the claim of general reorientation.”
>
> **Q: Not Claim general reorientation.**
>
> A: We will update the related text to clarify our considered task is only around the vertical axis, as well as the difficulty level compared to general in-hand reorientation. Please also refer to the Q2 of the shared response for a detailed discussion.
>
> ---
>
> > “The effective 2D aspect ratio of all objects is very close to 1. Although there are a few long objects present in the experiment, their cross sections (where the fingers are in contact) have an aspect ratio close to 1. This is okay as long as properly addressed in the limitation section. What would happen if the objects being manipulated (or their cross sections where the fingers are in contact with) have more extreme aspect ratios? What is limiting the object generalizability? It will be very interesting to see the extrinsics vector of a more extreme shape.”
>
> **Q: More Extreme Shapes.**
>
> A: We agree it is harder to rotate objects with extreme shapes. For example, the kiwi fruit has aspect ratio 1.5 and the performance is worse than regular shaped objects. We will include this discussion in our limitation section.

---

> ### Author Response · Authors · 2022-08-25
> **Follow up on the response**
>
> Thanks again for your time. We hope our previous response addressed your concerns. Please let us know if you have additional questions or need more clarifications that can help us improve the paper.

---

> ### Author Response · Authors · 2022-08-26
> **Appendix**
>
> We just realize viewing appendix is temporarily disabled, so we attached the appendix for your reference.

---

### Official Review · Reviewer_arnN · 2022-08-01

**Originality:** Poor
**Technical Quality:** Very Good
**Clarity Of Presentation:** Very Good
**Impact:** 3

**Recommendation:**

Weak Accept: I recommend accepting the paper, but will not argue for my recommendation if the majority of other reviewers have a different opinion.

**Summary:**

The paper employs Rapid Motor Adaptation to tackle the problem of policy generalization in in-hand manipulation tasks. The paper mainly describes results from a new experimental setup of in-hand manipulation. Generalization technique is along the similar lines of RMA, which tries to extract extrinsics vector from proprioceptive history of interaction with the world (environment). This extrinsics vector is hence used by a policy network to adapt its behavior to sensed changes in the environment.

**Issues:**

- Although the paper demonstrated a novel use case of an existing method, it doesn't present any new idea or new insights from an existing idea which is not ideal. It would be interesting if it could additionally comment on new insights gained from running the underlying method for a new experimental setup.
- It is not apparent to me how the policy behavior changes in response to a different environment (i.e., environmental variation). It would be helpful if this can be explicitly demonstrated as a video / plot. As an example, it would be interesting (at least to me) to see how the policy behavior changes with and without continuously using the adaptation module (e.g., by latching its output / using the same output as obtained from a different object).
- Could you also add qualitative results from the DR, SysID, NoAdapt, and Periodic baselines? In case I missed it, please point me to the correct videos.
- Lines (42-43): What type of gaits emerged from the baselines? Could you talk more about why having an adaption module could have led to smoother gaits?
- Lines (103-104): Could you explicitly states these changes?
- It would be greatly appreciated if you could expand on the limitations section.

I am looking forward to the discussion!

### Minor issues
- Lines (7-9): For a reader unaware of RMA, it can be slightly misleading what is being estimated by the method.
- Line (36): It has some grammatical issues. (Please also look at the remaining introduction section as it has additional grammatical errors.)
- Line (37): It is not clear to me what `adaptively estimate` means. Does it imply continuous updates?
- Line (56, 57): I am not sure, but: Starting with citation number might be aberrant with the template. You might want to double check.

**Quality Of The Limitations Section:**

Additional details required

**Reviewer Expertise:**

5: The reviewer is absolutely certain that the evaluation is correct and very familiar with the relevant literature

**Robotics Focus:**

Sufficient demonstration on hardware

**Strengths And Weaknesses:**

### Strengths
- The paper sufficiently motivates the problem statement.
- The paper presents an interesting application of an adaptation technique previously demonstrated for a different experimental setup.
- The qualitative hardware results seem sufficiently convincing and validate the applicability of the underlying method.
- The quantitive results (Figure 3) aptly explain how different methods respond to environmental perturbations.
- The presented experiments cover impressive variations in object shapes and dimensions.
- The ablation study is thorough and provides sufficient details about the training and testing procedures.

### Weaknesses
- Most importantly, the paper only presents a new application of an existing idea and lacks any new ideas.
- Furthermore, it does not present new insights on the existing idea (for instance, why having a separate extrinsic vector prediction network is a good design choice over directly training a meta-learning based policy with adaptive behavior).
- The paper and the underlying method’s adaptive functioning is limited by the ability to simulate environment/robot variations. It would be interesting if authors could provide any insights on method’s response to variations not previously seen during training. (For instance, by freezing one of the known variable during training and seeing how the method responds during testing.)

**Summary Of Recommendation:**

The paper presents a new and interesting experimental setup of an exisiting method but lacks originality in ideas or providing deep insights.

---

> ### Author Response · Authors · 2022-08-19
> **Re: Reviewer arnN**
>
> Thanks for your detailed review! We hope the following answers can address your concern.
>
> ---
>
> > Most importantly, the paper only presents a new application of an existing idea and lacks any new ideas.”
>
> **Q1: Research Contribution.**
>
> A1: Please see Q1/A1 of the shared response.
>
> ---
>
> > “Furthermore, it does not present new insights on the existing idea (for instance, why having a separate extrinsic vector prediction network is a good design choice over directly training a meta-learning based policy with adaptive behavior).”
>
> **Q2: New insights on the existing idea (why having a separate extrinsic vector and why not use meta-learning).**
>
> A2: We show the advantage of using a separate extrinsic vector in the appendix (Table 3). The DR-* removes the extrinsic vector prediction and directly uses proprioceptive history as input of the policy. We try an LSTM, and an MLP to process the input history. We found it is more difficult to optimize those policies, especially with longer input timesteps. We hypothesize that this is because jointly learning a universal policy for a large range of physical variations is difficult, while decoupling policy learning and extrinsic estimation into two stages makes it more effective.
>
> Using meta-learning for adaptation is another reasonable alternative, but it has not been demonstrated for this task yet. Meta-learning methods need reward values during deployment, which may need additional instrumentations such as vision sensing or markers on objects. In this project, we instead focus on exploring what a proprioceptive policy can achieve from onboard sensing. Moreover, we believe that meta-learning can complement our work on adaptation by improving the policy in the real world during deployment. We will discuss this aspect in the future work and limitations section in the updated version.
>
> ---
>
> > “The paper and the underlying method’s adaptive functioning is limited by the ability to simulate environment/robot variations. It would be interesting if authors could provide any insights on method’s response to variations not previously seen during training. (For instance, by freezing one of the known variable during training and seeing how the method responds during testing.)”
>
> **Q3: Method’s Response to Variations not Previously Seen during Training.**
>
> A3: We agree that several real-world variations cannot be accurately modeled in the simulation. However, what enables generalization to objects whose physical properties have not exactly been seen in simulation (objects with holes, shuttlecock) is that a seemingly different variation might yield an extrinsics vector similar to something seen during training.
>
> In Table 1 of the appendix, we quantitatively show the importance of simulating a large range of objects with diverse physical properties. We further study the effect of freezing only one of the properties (mass or scale) during training:
>
> | Method           | RotR         | TTF       | ObjVel    | Torque    |
> |------------------|--------------|-----------|-----------|-----------|
> | no randomization | 171.96±20.63 | 0.63±0.07 | 0.48±0.07 | 1.94±0.32 |
> | freeze scale     | 184.06±10.10 | 0.69±0.04 | 0.37±0.04 | 1.15±0.25 |
> | freeze mass      | 199.57±20.01 | 0.75±0.03 | 0.33±0.03 | 1.25±0.22 |
> | randomize all    | 222.27±21.20 | 0.82±0.02 | 0.29±0.05 | 1.20±0.19 |

---

> > ### Author Response · Authors · 2022-08-19
> > **Re: Reviewer arnN (Part 2)**
> >
> > > “Although the paper demonstrated a novel use case of an existing method, it doesn't present any new idea or new insights from an existing idea which is not ideal. It would be interesting if it could additionally comment on new insights gained from running the underlying method for a new experimental setup.”
> >
> > **Q4: New Insights Gained from Running the Underlying Method for a New Experimental Setup.**
> >
> > A4: Please see Q1/A1 of the shared response.. We conduct additional ablation experiments on the reward function as shown below.
> >
> > | Method                | RotR         | TTF       | ObjVel    | Torque    |
> > |-----------------------|--------------|-----------|-----------|-----------|
> > | remove pose penalty   | 131.37±10.84 | 0.85±0.04 | 0.41±0.03 | 2.05±1.06 |
> > | remove energy penalty | 212.33±10.28 | 0.79±0.02 | 0.48±0.02 | 1.73±0.22 |
> > | remove linvel penalty | 226.46±19.87 | 0.75±0.04 | 0.38±0.04 | 1.22±0.11 |
> > | full reward           | 222.27±21.20 | 0.82±0.02 | 0.29±0.05 | 1.20±0.19 |
> >
> > Without the pose penalty, the policy performs worse because the learned gait is unnatural and it does not learn to break and establish new contact. Please see an example in this link (http://anonymouscorl2022.github.io/ObjRotation/rebuttal.html#Reward). The energy penalty and linear velocity penalty greatly decrease the commanded torque and object’s linear velocity. This helps encourage the policy to learn a more stable and smooth gait to solve the task, and empirically yields better sim-to-real performance.
> >
> > We want to further clarify that we provide insights for this task in Figure 5, Figure 6, and the whole section 5.2. We qualitatively show that the inferred mass and scale properties correlate with the real world values of the object. We also show that we can cluster different objects based on their extrinsics, and we qualitatively see objects with similar properties clustered together, showing the benefit of a compressive embedding.
> >
> > ---
> >
> > > “It is not apparent to me how the policy behavior changes in response to a different environment (i.e., environmental variation). It would be helpful if this can be explicitly demonstrated as a video / plot. As an example, it would be interesting (at least to me) to see how the policy behavior changes with and without continuously using the adaptation module (e.g., by latching its output / using the same output as obtained from a different object).”
> >
> > **Q5: How policy changes in response to a different environment. Videos showing how policy changes with and without continuously using the adaptation module.**
> >
> > A5: In Figure 5 and video of the supplementary material, we change the rotated objects during policy operation and show that the predicted extrinsics change when the object changes. The policy continuously updates the extrinsics at 20Hz. Following your suggestion, we do another experiment where we freeze the extrinsics and then change the object during rotation. Please use see the results in this link (https://anonymouscorl2022.github.io/ObjRotation/rebuttal.html#R1Q1).
> >
> > ---
> >
> > > “Could you also add qualitative results from the DR, SysID, NoAdapt, and Periodic baselines? In case I missed it, please point me to the correct videos.”
> >
> > **Q6: Qualitative results from the DR, SysID, NoAdapt, and Periodic baselines.**
> >
> > A6: Thanks for the suggestion, we have added it to our website (https://anonymouscorl2022.github.io/ObjRotation/rebuttal.html#R1Q2).

---

> > > ### Author Response · Authors · 2022-08-19
> > > **Re: Reviewer arnN (Part 3)**
> > >
> > > > “Lines (42-43): What type of gaits emerged from the baselines? Could you talk more about why having an adaption module could have led to smoother gaits?”
> > >
> > > **Q7: Lines (42-43): What type of gaits emerged from the baselines? Could you talk more about why having an adaption module could have led to smoother gaits?**
> > >
> > > A7: The baselines have a similar gait as ours because they are trained with the same reward function and under the same environment (as also shown in the updated baseline videos). The reward function we use encourages the policy to have a smooth gait by penalizing the applied torque and energy consumed. This coupled with the training setup is what leads the policy to choose the specific gait suitable for the task.
> > >
> > > The role of the adaptation module is to allow the policy to determine which object is being dealt with, and apply only as much torque as is necessary to successfully rotate the object. In the absence of the adaptation module, the policy would be forced to use a high torque and a conservative gait as we show with the DR baseline in Figure 3, Figure 4 in the paper. We further show the importance of the training environment in emergent gaits in the supplementary material (Section B, The Different Emergent Gaits between using Cylinder or Sphere for Training) and website at submission time. We will update the main paper and make this point clearer in the experimental section.
> > >
> > > ---
> > >
> > > > “Lines (103-104): Could you explicitly state these changes?”
> > >
> > > **Q8: Lines (103-104): Could you explicitly state these changes?**
> > >
> > > A8: The reward and the training environments need to be designed for this task. Please refer to Q1 of the shared response and the above Q4/A4. We will explicitly state these in the revision.
> > >
> > > ---
> > >
> > > > “It would be greatly appreciated if you could expand on the limitations section.”
> > >
> > > **Q9: Expansion of the Limitation Section.**
> > >
> > > A9: Thanks for your suggestion. We will adjust the length of the other sections and expand the limitation section. We will explicitly say the possibility of using meta-learning for improving our policy during deployment, clarify the difficulty of the task, and detail the set of objects that our policy fails to deal with.
> > >
> > > ---
> > >
> > > > “Lines (7-9): For a reader unaware of RMA, it can be slightly misleading what is being estimated by the method.
> > > Line (36): It has some grammatical issues. (Please also look at the remaining introduction section as it has additional grammatical errors.)
> > > Line (37): It is not clear to me what adaptively estimate means. Does it imply continuous updates?
> > > Line (56, 57): I am not sure, but: Starting with citation number might be aberrant with the template. You might want to double check.”
> > >
> > > **Q10: Minor Issues.**
> > >
> > > A10: Thanks for your suggestion. Line (37) should be changed to “continuously” instead of “adaptively”. We will proof-read the text and correct  it in the updated pdf file.

---

> ### Author Response · Authors · 2022-08-25
> **Follow up on the response**
>
> Thanks again for your time. We hope our previous response addressed your concerns. Please let us know if you have additional questions or need more clarifications that can help us improve the paper.

---

> ### Author Response · Authors · 2022-08-26
> **Appendix**
>
> We just realize viewing appendix is temporarily disabled, so we attached the appendix for your reference.

---

### Author Response · Authors · 2022-08-19
**Shared Response to All Reviewers**

**Q1: Novelty and Scientific Contribution.**

A1: We want to respectfully emphasize that in-hand object rotation with  a diverse set of objects varying in mass, softness, and shapes remains to be an open challenge for the robotics community. We believe solving this challenge itself, by any method, is a significant contribution. On the other hand, the critique of “just an application of an existing idea” would be valid if the “existing idea” (RMA) was a standard, well-known technique that has already been applied in multiple domains. But this is not true for RMA. It was presented very recently (2021), and its application has only been in legged locomotion, a very different domain, and by one research group. It is not at all obvious whether it would work for this very different and challenging task. In fact, to make it work for the in-hand object rotation task, we needed at least two innovations:

1) Reward: One of the criticisms of RL for robotics is that complex hand designed reward functions are needed. We found a natural reward function without having to pre-specify contact patterns. This was enough for the emergence of natural gaits, which could then be transferred to hardware.
2) Training environment: A good training environment has to provide enough variety in simulation to enable generalization in the real world. RMA in the walking context used fractal terrains. In this work we find that using cylinders with different aspect ratios and masses provides such variety, whereas spheres do not.

We believe our simple and effective solution to the in-hand rotation task is an important step towards the goal of general in-hand manipulation and interaction with real-world objects.

---

**Q2: Clarification on the Task Difficulty**

A2: We agree that there are different levels of difficulty for general dexterous in-hand manipulation. The easiest one would be rotating objects in a supporting surface such as a palm or a table, which is a 2D planar rotation task. The final goal would be a system that can do full SO(3) general object reorientation. The task of in-hand rotation which we consider in this paper is more difficult than 2D planar rotation because our policy does not rely on a supporting surface to maintain stability. In the submission, we’ve tried to reflect this in the title of our paper – “object rotation”, instead of “general in-hand manipulation”. We also state in the abstract that our work is a “small step” towards that.

Note that, although we simplify the general SO(3) problem to the task of in-hand rotation, it is not a limiting simplification. With three policies for rotation along different axes (x, y, and z), we can achieve rotating the object to any target pose. We view our task as an important step towards the more difficult general object in-hand reorientation, and we will state the difficulty level of the task more clearly in the revision.

---

### Meta-Review · Area_Chair_yvgx · 2022-08-31

**Recommendation:** Accept (Poster)
**Confidence:** 3

**Metareview:**

This paper describes a system that is trained in simulation to manipulate cylindrical objects. Then it generalizes to manipulate a variety of objects in the real world with no fine-tuning. It uses a low-dimensional representation of the object shape as perceived by the robot's finger tips to achieve this.

Strengths:

Addressing an important problem of dexterous manipulation.
Concrete results on a real physical robot, performing sim-to-real transfer.
Weaknesses:

More extreme shapes with larger aspect ratios are harder to rotate.
Rotation is only in two dimensions and manipulation is quasistatic; the limitations of the approach need to be clearly addressed in the paper.